# Transfer Learning for Automatic Sleep Staging Using a Pre-Gelled Electrode Grid

**DOI:** 10.3390/diagnostics14090909

**Published:** 2024-04-26

**Authors:** Fabian A. Radke, Carlos F. da Silva Souto, Wiebke Pätzold, Karen Insa Wolf

**Affiliations:** Fraunhofer Institute for Digital Media Technology IDMT, Oldenburg Branch for Hearing, Speech and Audio Technology HSA, 26129 Oldenburg, Germany

**Keywords:** sleep staging, electrode grid, EEG, machine learning, transfer learning, home monitoring

## Abstract

Novel sensor solutions for sleep monitoring at home could alleviate bottlenecks in sleep medical care as well as enable selective or continuous observation over long periods of time and contribute to new insights in sleep medicine and beyond. Since especially in the latter case the sensor data differ strongly in signal, number and extent of sensors from the classical polysomnography (PSG) sensor technology, an automatic evaluation is essential for the application. However, the training of an automatic algorithm is complicated by the fact that the development phase of the new sensor technology, extensive comparative measurements with standardized reference systems, is often not possible and therefore only small datasets are available. In order to circumvent high system-specific training data requirements, we employ pre-training on large datasets with finetuning on small datasets of new sensor technology to enable automatic sleep phase detection for small test series. By pre-training on publicly available PSG datasets and finetuning on 12 nights recorded with new sensor technology based on a pre-gelled electrode grid to capture electroencephalography (EEG), electrooculography (EOG) and electromyography (EMG), an $F_{1}$ score across all sleep phases of 0.81 is achieved (wake 0.84, N1 0.62, N2 0.81, N3 0.87, REM 0.88), using only EEG and EOG. The analysis additionally considers the spatial distribution of the channels and an approach to approximate classical electrode positions based on specific linear combinations of the new sensor grid channels.

## 1. Introduction

Sleep stage classification is widely used for diagnostics for sleep-related disorders. Polysomnography (PSG) is used to diagnose such disorders. For PSG, patients spend one night in a sleep laboratory where electroencephalography (EEG) and other modalities are recorded. While disorders like sleep apnea can be diagnosed that way, there are multiple sleep disorders or sleep-related sleep disorders like insomnia that demand multiple recordings over a longer time span [1]. Therefore, it is not practical to make recordings for these patients, but it is more useful to monitor sleep at home, which requires self-administered EEG systems. Such systems also enable a long-term monitoring for detection of sleep-related disorders like Parkinson’s and Alzheimer’s disease [2]. Additionally, the PSG introduces inconvenience for the patients, as they need to come into a sleep laboratory and the measurement setup can have a bad influence on the patients’ sleep. Certainly, the unfamiliar environment further worsens the sleep during the recording, known as the first night effect [3]. The gold standard for staging the recording is manual scoring by sleep clinicians according to AASM (American Academy of Sleep Medicine) or Rechtschaffen and Kales [4,5] guidelines, recorded at a sleep laboratory. However, this scoring process is time-consuming and prone to inter-rater variability [6]. To overcome these issues and improve the care of patient suffering sleep disorders, two solution approaches can be identified: addressing the environment issue, we propose the usage of an EEG setup for recording, that is self-applicable and allows EEG sleep monitoring at home at a high quality.

Different mobile solutions have been and are being developed to achieve an easy-to-use home monitoring system: the wet electrode grids cEEGrid [7,8] and trEEGrid [9], dry electrode solution like Dreem [10,11] or the pre-gelled HomeSleepTest [12]. The trEEGrid covers a wider range of electrode positions than the cEEGrid, see Figure 1, and it aims to be less intrusive and easier to use due to pre-gelled electrodes. The gel is used to obtain a better signal than with dry electrodes while at the same time being more comfortable to wear. It is shown based on a conceptual prototype of the trEEGrid that high quality hypnograms can be derived [9].

As in many other fields of health care, the application of artificial intelligence has become feasable for automatic sleep staging. Various methods using deep neural networks have achieved annotation quality in the automatic sleep staging task that is comparable to the quality achieved by humans [14]. Most popular are sequence-to-sequence architectures made of epoch and sequence encoder structure [15,16,17], often consisting of recurrent neural networks (RNNs) such as long-short-term memory (LSTM) [18] and gated recurrent units (GRUs) [19]. Also fully convolutional networks [20], U-nets [21,22] and transformer architectures [23] have been proposed for automatic sleep staging. These methods use large databases, usually recorded with a classical PSG setup, of which most models utilize only a small number of channels. The use of these large databases is not suitable or possible in the case of specialized EEG setups, as the electrode setup differs both in the number of available channels and electrode positioning. This creates a domain mismatch. Application of transfer learning is a method for addressing the issue of domain mismatch in various fields, including health care, when artificial intelligence is employed [24]. This is also the case for automatic sleep staging, where different approaches to transfer learning were investigated [25]. To accommodate for channel mismatch between datasets for neural networks that predict sleep staging on a single channel, transfer learning was used successfully [26,27,28]. To deal with this and other domain mismatch issues, such as differences in the demographics between the training and inference data, transfer learning can be applied to automatic sleep staging [29,30]. In [30], a sequence-to-sequence architecture for automatic sleep staging is proposed and evaluated for transfer learning across different databases with varying electrode setups. In [17,31], the authors show the application of transfer learning for cEEGrid. Besides domain mismatch, the use of transfer learning was used to investigate the personalization of sleep staging algorithms [32,33].

This study evaluates the usage of the trEEGrid for automatic sleep staging and additionally investigates the influence of channel combinations to calculate virtual electrode positions for this use case. Our main contributions are as follows: First, we apply sleep staging algorithms to recordings made with the trEEGrid. Second, we investigate the utilization of channel combinations for automatic sleep staging, which has been shown to be helpful for sleep expert scorers following a linear combination model to approximate classical electrode positions [9]. Thirdly, we use transfer learning, with recordings made with the trEEGrid as the target domain. Furthermore, we investigate the influence of channel combinations on the transfer learning for the trEEGrid. Experimental results demonstrate the applicability of the proposed method to automatic sleep staging aimed at at-home sleep monitoring applications in the future. We have limited ourselves here initially to data from healthy individuals in order to keep the analyses and results straightforward. For practical application, the transfer to sleep data with different pathologies is important in a further step.

## 2. Methods

Sleep staging is the task of assigning one of the five sleep stages “wake” (W), “rapid-eye-movement” (REM), “non-REM 1” (N1), “non-REM 2” (N2), “non-REM 3” (N3) defined by the AASM rules to segments of a PSG recording consisting of EEG and additional further sensors during sleep. The mentioned segments are usually 30 s long due to historical reasons. Throughout this work, we use only EEG and electrooculgraphy (EOG) signals of a subject’s recorded night. The signals used in this study consist of multiple channels, containing voltages between a common reference electrode and the signal electrodes derived from the scalp.

To investigate the capabilities of the novel trEEGrid which provides EEG and EOG signals, we perform the task of automatic sleep staging on different sets of recordings. In the following section, we describe the source dataset used for transfer learning and the dataset recorded with the trEEGrid, as well the network architecture used to perform the automatic sleep staging and the metrics to quantify the resulting performance.

### 2.1. Data

We use multiple datasets in this study to investigate the usage of transfer learning for data recorded with the trEEGrid. As the source domain, we use the Montreal archive of sleep studies (MASS), cf. [34]. Recordings made with the trEEGrid represent the target domain in the experiment. Those datasets and the pre-processing are described in the following.

#### 2.1.1. Montreal Archive of Sleep Studies

The MASS database of which the first cohort consists of five subsets was recorded in the center for Advanced Research in Sleep Medicine, Montreal, Canada. In this study, we utilize the third subset (SS3) to perform our experiments. SS3 consists of 30 s epochs, which were scored according to the AASM standards. The dataset consists of 62 recordings, each from 62 different healthy subjects, which accumulate to a total recording time of 21.8 days. The subject’s mean age is 42.5 years, with a standard deviation of 18.9 years. A total of 55% of the subjects are female and 45% are male. Recordings were made with an full PSG montage, of which we use several EEG and EOG channels. EEG channels available for all 62 subjects in SS3 are Fp1, Fp2, Fz, F3, F4, F7, F8, Cz, C3, C4, Pz, P3, P4, T3, T4, T5, T6, Oz, O1 and O2. Left and right horizontal EOG channels are also included. The recordings are sampled at a sampling rate of 100 Hz.

#### 2.1.2. trEEGrid Database

This database was recorded to evaluate a prototype of a self-applicable grid that is able to record EEG, EOG and EMG data, cf. [9] and Figure 1. There are 20 datasets recorded with the trEEGrid prototype in combination with a wireless EEG amplifier (Smarting Sleep, mBrainTrain, Belgrade, Serbia) from young adults (mean age = 28.9 years) who reported to have no sleep disorders. In parallel to the recording with the trEEGrid prototype, a classical PSG recording was made simultaneously with a commercial mobile PSG system (SOMNOscreen Plus, SomnoMedics, Randersacker, Germany). The commercial system included six EEG electrodes, two piezosensoric belts for thorax and abdomen expansion and a finger clip sensor, which measured oxygen saturation and pulse. The two EMG electrodes of the PSG system were not used, since they competed with the space of the trEEGrid EMG channels R6 and R7. PSG electrodes were gold-plated cup electrodes mounted with adhesive gel.

Eight recordings had to be excluded due to technical problems either with the trEEGrid (six), with the Bluetooth connection to the amplifier (one) and with the PSG system caused by a power failure (one) so that 12 datasets remain for the sleep staging computations.

Both recordings were scored by an experienced sleep expert scorer according to the AASM rules. To perform the experiments, we use the labels made on the basis of the classical PSG recording. There is a total of five epochs in the 12 recordings, that could not have a label assigned, due to artifacts. The electrodes and the channels, respectively, are numbered from R1 to R7, as depicted in Figure 2. Channels R6 and R7 are aimed at EMG acquisition and will be ignored in the rest of this paper. Recordings are made with a sampling rate of 250 Hz. Recorded signals were bandstop filtered from 45 to 55 Hz and 60 to 65 Hz to reduce line noise and impedance current. Impedance current was used through the measurements to record impedance continuously for evaluation of the electrode grid prototype. Furthermore, a bandpass filter from 0.5 to 40 Hz was applied to the signals. All filters used are phase true Butterworth filters of fourth order.

In [9], the benefit of linear combinations of different channels as approximation of classical electrode positions was investigated. The specific linear combinations are sketched in Figure 2. The same approach using re-referenciation is used here to approximate the signal of the standard PSG setup based on the signals recorded with the trEEGrid.

### 2.2. Preprocessing

To align the recordings with each other, we performed resampling to a sampling rate of 60 Hz. Before the resampling, re-referenciation may be performed. After that, we perform interquartile range normalization for each channel independently as follows
(1)$$\begin{array}{c} x_{k}^{norm}=\frac{x_{k}}{\mathrm{IQR}(x_{k})} \end{array}$$
where x is a vector containing t∈1,…,T samples and IQR(·) is an operator that calculates the interquartile range of the signal x for each channel *k* along the time *t*. Sections in the signal that exceed the interquartile range by twenty times were hard clipped within this range.

### 2.3. Model Architecture

In this work, we use the RobustSleepNet architecture proposed in [30], which is a variant of the sequence-to-sequence structure often used for automatic sleep staging. Transfer learning has been applied to this architecture, as it tackles the problem of electrode mismatch between datasets. The use of an attention mechanism to aggregate across the input channels enables the architecture to work efficiently in transfer learning with a different number of channels.

The network shown in Figure 3 is structured as follows: an epoch encoder, extracting a feature vector representing each epoch, a sequence encoder processing temporal dependencies among the sequence of epochs and finally a classification layer. The epoch encoder first normalizes the input features by mean and variance. Next, the RobustSleepNet employs a linear layer to reduce the frequency dimension. This linear layer shares the same weights across all channels. All input channels are then recombined using multi-head attention, so that the subsequent recurrent network receives a constant number of input channels. This two-layer bidirectional GRU network, operating along the time dimension, processes the information distributed within each epoch. Dropout is employed before and after the GRU network. To aggregate the GRU’s output into a single vector representation for each epoch, an attention layer that accumulates information along the time dimension is employed.

The sequence encoder receives the feature vectors that were computed by the epoch encoder for each epoch and aims to model the inter-epoch dependencies. It consists of a bidirectional GRU network with two layers with skip connections. Skip connections are implemented by a linear layer, whose transformed representations are added to the output of the associated GRU layer. Also, after each GRU layer, dropout is employed. The sequence encoder consisting of GRU layers models the inter-epoch dependencies in the sleep explicitly.

The classification scores are calculated based on the output of the sequence encoder. There is a linear layer with an output dimension of five, followed by a softmax activation to achieve log likelihoods for each class. Each output dimension represents one out of the five sleep stages.

We use the network with the same parameters as described in [30]. The network consists of approximately 180 k trainable parameters.

### 2.4. Loss Function

We employ the cross entropy loss function defined by
(2)$$\begin{array}{c} L\left(x,y\right)=\frac{1}{N}\sum_{n=1}^{N}\sum_{c=1}^{C}x_{n,c}y_{n,c} \end{array}$$
where $x_{n,c}$ denoted the log-likelihood and $y_{n,c}$ the label for class *c*, with c∈[1,C] and epoch *n*. *C* represents the number of classes, which is five. Epochs that are not labeled as one of the sleep stages due to artifacts do not have a class label c∈[1,C], so these epochs are excluded from loss computation. Each batch consists of *N* epochs, the product of batch size and context size.

### 2.5. Inference

During inference time, we sample a whole night, so that we concatenate all epochs into one sequence. This way, the RNN in the sequence encoder works along the whole sequence. This differs from the scoring method described in [30], in which a shifting context sequence of length 21 was scored, with a stride of one and geometric averaging is employed on the scores across all classification of an epoch. We changed the inference method, as we found no improvement caused by this ensemble scoring method, despite the mismatch between training and inference setup.

### 2.6. Channel Setups

As described before, the dataset that we use to train and test the model were recorded on different EEG montages. To adapt the montages to each other, we choose to select the EEG channels in a way that the recorded signals are more similar. In particular, we take five different setups into consideration, three setups from the source domain (SS3, SS3^grid^, SS3^grid-PSG^) and two from the target domain (trEEGrid, trEEGrid^PSG^). While the SS3 setup takes all available classical PSG EEG and EOG channels into account, the other setups aim to match the signals in the other domain, respectively. All setups are listed in Table 1. In the SS3^grid-PSG^ setup, classical electrode positions close to the electrode positions realized with the novel trEEGrid sensor were chosen. The trEEGrid setup uses all available channels of the trEEGrid sensor grid with the original reference close to the mastoid. For the trEEGrid^PSG^ setup, R1 and R5 are used, being close to the classical position Fpz and C4. Additionally, re-referencing was applied to approximate classical EEG electrode position O2, as sketched in Figure 2. In SS3^grid^, we take a subset of MASS SS3 channels into consideration that corresponds to the channels approximated by the trEEGrid^PSG^ setup. As the MASS SS3 dataset does not contain an Fpz channel, it is approximated by the average of the channels Fp1 and Fp2. All setups based on the trEEGrid dataset also include a diagonal EOG channel. As this is not available in MASS SS3, we use the right horizontal EOG channel in there instead.

## 3. Experiments

### 3.1. Training Procedure

During training, the batch size was 12. Each training example of the batch consists of 21 consecutive epochs, resulting in 252 epochs in a training batch in total. To deal with the class imbalance problem, epochs were sampled in a weighted manner, similar to [21]. This means that, for each sequence in a batch, we first draw one epoch from the dataset. This epoch is drawn randomly, but weighted, to counteract the class imbalance. The drawn epoch is then given a random position in the sequence, so that the context is filled with the adjacent epochs. Some of the adjacent epochs will represent a different sleep stage than the balanced drawn epoch; nevertheless, this sampling method still mitigates the imbalance problem. One training iteration over the dataset is considered as finished, when there were as many training examples drawn as there are epochs in that dataset. Due to this method, iterating over the dataset does not necessarily contain all samples in one iteration. Nevertheless we did not ensure that an iteration contains every sample at least once and left the length of one iteration to the number of epochs in the dataset.

Cross validation was used to train on the target datasets, as well for training on the source dataset. The datasets were split so that each subject’s epochs were present in only one of the training, validation or test set. For MASS SS3, 10-fold cross validation was used. As the trEEGrid dataset consists only of 12 recorded nights, we use a leave-one-night-out scheme, resulting in 12 cross validation folds. The sizes of training, validation and test sets for the cross validation as well as the number of folds are depicted in Table 2. Percentages for the SS3 dataset are target proportions, which are varied across folds to accommodate that the total number of recordings is not divisible by the number of folds.

We implemented the experiments using PyTorch in version 1.12.1 and pytorch-lightning in version 1.7.7. As an optimizer, Adam [35] with parameters (learning rate: $10^{-4}$, $\beta_{1}$: 0.9, $\beta_{2}$: 0.999) and weight decay regularization (weight decay: $10^{-3}$) was used. All weights and biases of the model layers were initialized with values drawn from a random uniform distribution with upper and lower bounds $\left(-\sqrt{l},\sqrt{l}\right)$, where *l* for each layer is defined as $l=\frac{1}{\mathrm{number}\;\mathrm{of}\;\mathrm{input}\;\mathrm{features}}$. The models were trained on an NVIDIA Quadro RTX 8000 GPU, on which a 10-fold cross validation run on the MASS SS3 subset takes approximately 3 h. We use early stopping as we end training after the validation loss did not decrease for 25 training iterations. Training was stopped after a maximum of 100 training iterations. The same seed to initialize all random number generators was used for all runs.

### 3.2. Transfer between Domains

For investigation, trainings on the source domain (MASS dataset) and target domain (trEEGrid) were performed. We show the learning-from-scratch (LFS) performance for both, the source and target domain, where the cross validation and the datasets was performed. The direct transferability between domains is quantified by applying a model trained on the source domain directly on the target domain, without any adaption to the new domain. We denote this as direct transfer (DT). Finally, we report the transfer learning performance, for which a mode, trained on the source domain data is finetuned (FT) on the target domain data. In this case, the weights for the sequence encoder and the classifier were fixed, so that only the epoch encoder becomes finetuned for the target domain. For DT and FT, the model that was trained on the source dataset and showed the lowest validation loss during cross validation was used.

## 4. Performance Metrics

To determine the algorithm’s performance, we use the $F_{1}$ scores as well as Cohen’s Kappa. The scores are always calculated across all folds of the cross validation. This is carried out by first calculating the $F_{1}$ scores and Cohen’s Kappa on the test data for every fold. Then the mean and the standard deviation is determined across all folds for each metric.

### 4.1. $F_{1}$ Score

Reported $F_{1}$ scores for each class were calculated from the number of true positive (TP), false positive (FP) and false negative (FN) for each class, as
(3)$$\begin{array}{c} F_{1}=2\frac{\mathrm{precision}\cdot \mathrm{recall}}{\mathrm{precision}+\mathrm{recall}}, \end{array}$$
with $\mathrm{precision}=\frac{\mathrm{TP}}{\mathrm{TP}\;+\;\mathrm{FP}}$ and $\mathrm{recall}=\frac{\mathrm{TP}}{\mathrm{TP}\;+\;\mathrm{FN}}$. $F_{1,c}$ denotes the $F_{1}$ score for class *c*. The macro-$F_{1}$ score which gives a measure across all *C* sleep stages was obtained by averaging over the $F_{1}$ score of each class:(4)$$\begin{array}{c} \mathrm{Macro}-F_{1}=MF_{1}=\sum_{c=1}^{C}F_{1,c} \end{array}$$

### 4.2. Cohen’s Kappa

Cohen’s Kappa is often used to measure the inter-rater agreement in the context of medical annotations. As we compare the performance of the automatic sleep staging to the human achievable performance, we also calculated Cohen’s Kappa κ as follows:(5)$$\begin{array}{c} \kappa =\frac{p_{o}-p_{e}}{1-p_{e}}. \end{array}$$
$p_{o}$ stands for the observed probability that two related labels correspond to the same class, while $p_{e}$ is the expected probability when randomly assigning labels. Cohen’s Kappa was calculated across all sleep stages, denoted as κ.

## 5. Results

The automatic sleep staging was applied to the five different channel setups, of which the setups using the MASS SS3 dataset act as the source domain in the case of transfer learning. In Table 1, scores for LFS, DT and FT as well as the inter-scorer variability of our scorer [9] on the trEEGrid dataset and according to [6] are listed.

Because the number of datasets has a large impact on classification performance in addition to differences in electrode montage, we conducted a preliminary examination of the effect of the number of nights used in the modeling. For that, we randomly sampled subjects of the MASS SS3 database to create artificially smaller datasets, starting with 10 subjects, going up to the whole MASS SS3 dataset. For each number of subjects, 20 trainings were performed on newly drawn subsets to asses the variation between runs. Each training consists of one split with again 70% train data, 20% validation data and 10% test data. We report the median macro-$F_{1}$ over the 20 repeated runs and its 25 and 75 percentiles. As shown in Figure 4, having a smaller dataset decreases the performance achieved by the trained network, as well as the variety in performance across multiple iterations. It can be seen that a dataset consisting of subjects in the low tens leads to a median macro-$F_{1}$ score of around 0.75. From a number of around 40 datasets, a macro-$F_{1}$ well above 0.80 is achieved.

To asses in a second step the differences between the five setups, the LFS performance was calculated. Next, the performance of transfer learning from the SS3 setups to the matching trEEGrid setups was calculated. Performance metrics are listed in Table 3.

In the LFS scores, it can be seen that the algorithm benefits from multiple channels present in the SS3 dataset. Reducing the channels for the SS3^grid^ and SS3^grid-PSG^ setups deteriorates the performance slightly, compared to the SS3 setup. Automatic sleep staging performs worse on both trEEGrid setups than on the SS3 setups, which can be expected due to the smaller dataset. Here, the trEEGrid setup that uses the default trEEGrid channels shows worse performance in both Cohen’s Kappa and macro-$F_{1}$ scores than trEEGrid^PSG^ setup, which has less channels in total.

The DT condition reveals the domain mismatch between the source and respective target datasets. So in direct transfer the average macro-$F_{1}$ score is in all setups in a range between 0.63 and 0.65 and worse than the LFS situation. This indicates that none of the models based on SS3 datasets succeed in the direct transfer to the new domain compared to the results achieved with the LFS approach even on small datasets. However, when considering Cohen’s Kappa, models trained on a subset of available channels of MASS SS3 (SS3^grid^, SS3^grid-PSG^) tend to perform better than their counterparts trained on SS3.

In the case of FT condition, in every case the performance improves compared to DT, in both Cohen’s Kappa and macro-$F_{1}$ score. More improvement is observed when SS3^grid^ and SS3^grid-PSG^ were used for pre-training. As in LFS, the trEEGrid^PSG^ setup as target domain tends to outperform the trEEGrid setup. This is even the case if pre-training happened on SS3^grid^, where larger channel mismatch is expected. Compared to the results of DT, it can be observed for FT also in macro-$F_{1}$ score that the pre-training on SS3 performs worse on both target domains, trEEGrid and trEEGrid^PSG^. Thus, a more generalized model in terms of input channels does not give an advantage here. Finetuning to the trEEGrid^PSG^ from both SS3^grid^ and SS3^grid-PSG^ performs best, with almost equal performance in terms of Cohen’s Kappa and macro-$F_{1}$ score. In these cases, the automatic sleep staging performs better on the data recorded with the trEEGrid than from a human scorer on the same data [9] and in terms of Cohen’s Kappa it achieves an accuracy as scorer on classical recordings following the AASM rules.

## 6. Discussion

This study investigates the use of transfer learning for sleep staging based on a small dataset of only 12 recordings obtained with a prototype of a self-applied electrode grid, so-called trEEGrid. We trained RobustSleepNet [30], an established neural network for this task, on both the MASS SS3 database and the dataset recorded with the trEEGrid prototype. The influence of the channel setup in both domains on the performance during the transfer was investigated. Also, the influence of the size of the dataset on the expected performance and the uncertainty was considered. This demonstrated a deterioration in the system performance with a reduction in the number of records in the dataset, accompanied by a higher uncertainty between multiple runs. Sources for this uncertainty are the random components in the training procedure, such as the random initialization of the trainable network parameters, the random selection of training records from the MASS SS3 dataset and the random order in which training samples are compiled in batches during the training process.

For smaller datasets, a decreasing performance and a higher uncertainty are generally to be expected. However, the results show that it also depends on the selected channels and their combination. In the results, it can be seen that LFS scores are highest for the SS3 setup, the largest dataset. SS3^grid^ and SS3^grid-PSG^ show worse performance, which can be explained by the fact that these setups have fewer channels. However, if the variance is taken into account, the three setups based on MASS S3 are very similar to each other. Reported performance measures are always influenced by the random nature of the training procedures of neural networks, so that differences must not be over-interpreted and the variance must be taken into account.

Both trEEGrid datasets have also a limited number of channels but additionally a much smaller number of records (12 nights). The dataset trEEGrid achieves the worst result of the LFS runs. However, by combining the channels with the aim of approximating classic electrode positions, better results are achieved with trEEGrid^PSG^, even though one channel less is included in the model. By adding the linear combination R5-R4, which approximates the classical channel O2 in the trEEGrid^PSG^ setup, new independent information is available to the model, compared to the information of channel R1, R2, R4 and R5 where (R1 and R2) and (R4 and R5) have a similar orientation and thus similar information. A more specific investigation of this form of signal approximation has been carried out in relation to sleep spindles [9]. The larger variance values for the performance data must be taken into account here as well.

For DT, the results show worse performance, which is expected due to domain mismatch. However, the matching channel setups SS3^grid^ to trEEGrid and SS3^grid-PSG^ to trEEGrid^PSG^ do not help in direct transfer. This hints at a still remaining channel mismatch, which can arise due to differences in the exact positions between the trEEGrid electrodes and the EEG cap used in the recording of the MASS SS3 database. Although the channels for SS3^grid^ and SS3^grid-PSG^ were selected to correspond with the channels represented in trEEGrid and trEEGrid^PSG^ setups, respectively, these channels remain distinct in their placement and, consequently, in their recorded signal.

The FT for the trEEGrid datasets as the target domain shows no improvement over the LFS if the base dataset SS3 is used as source domain. The performance for the target domain trEEGrid also remains limited, even if the sub-datasets SS3^grid^ or SS3^grid-PSG^ are used as source domain. A performance gain is achieved if trEEGrid^PSG^ is selected as the target domain. Here, for both source domains SS3^grid^ and SS3^grid-PSG^ finetuning to trEEGrid^PSG^ results in highest scores in terms of both Cohen’s Kappa and $F_{1}$ metrics. Finetuning on the trEEGrid dataset improves the performance relative to LFS and DT, but cannot achieve the same performance as trEEGrid^PSG^. Again, this is a hint concerning the benefit of using the linear combination R5-R4.

It is important to note that while the approach is promising, its current application in healthcare settings may be limited because the datasets used consisted exclusively of healthy participants. Sleep architecture can vary widely due to a number of factors, including age and various sleep disorders. Therefore, using only data from healthy individuals may not provide a sufficient basis for clinical applications. For example, it has been shown that a model trained on recordings from healthy participants suffers when applied to patients with obstructive sleep apnea because this disease pattern results in higher sleep fragmentation [11]. It would be beneficial to include assessments using sleep data from patients with different pathologies. Depending on the prevalence of the specific pathology, only a small number of recordings may be available; the transfer learning approach could also be advantageous here.

## 7. Conclusions

Using a sequence-to-sequence network, we showed the performance of a sleep staging algorithm finetuned on a small dataset of recordings made with the trEEGrid, a prototype of a self-applicable electrode grid capable of recording EEG, EOG and EMG at home. The used network architecture, previously used for transfer learning on multiple larger datasets is applied here on a smaller dataset. We can see that finetuning helps to improve the performance on this very small dataset, consisting of 12 records. Furthermore, it is demonstrated that the approximation of the source dataset towards the target dataset helps to improve the performance for finetuning. This is relevant not only in the use of automated sleep staging for new sensor technology for which there is a limited amount of observational data possible, but also for special disease conditions for which few datasets are available. The datasets used in this study are from healthy adults. Further analysis needs to be conducted on the application to data with different sleep pathologies. Transfer learning could also be a way of improving model performance here. Finally, transfer learning based on small datasets of the target domain is of interest for classification models that are trained individually for specific subjects to enable long-term observation. Our results also indicate that while a more generalized pre-training can aid in the direct transfer of a trained model to a new domain, it does not improve the preconditions when finetuning is necessary. We demonstrated that using re-referenced channels to estimate signals that are not directly measurable with the trEEGrid improves automatic sleep staging, particularly when trEEGrid recording is utilized to approximate classical PSG setups. As a result, automatic sleep staging outperforms human sleep scorers when scoring the same data and performs on par with human scorers on classical PSG recordings.

## Figures and Tables

**Figure 1 diagnostics-14-00909-f001:**
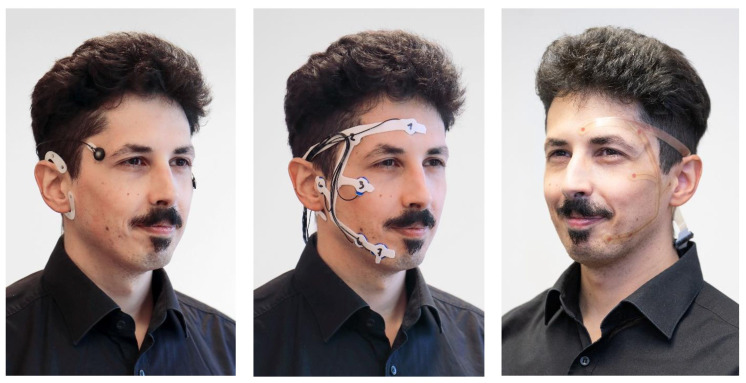
Development steps of the self-applicable, pre-gelled trEEGrid. **Left**: cEEGrid + EOG [13]. Middle: foam trEEGrid with labeled channels [9], used for recordings in this study. **Right**: trEEGrid prototype on a flexible circuit board. ©Fraunhofer IDMT/Anika Bödecker. Figure and caption are adapted from [9].

**Figure 2 diagnostics-14-00909-f002:**
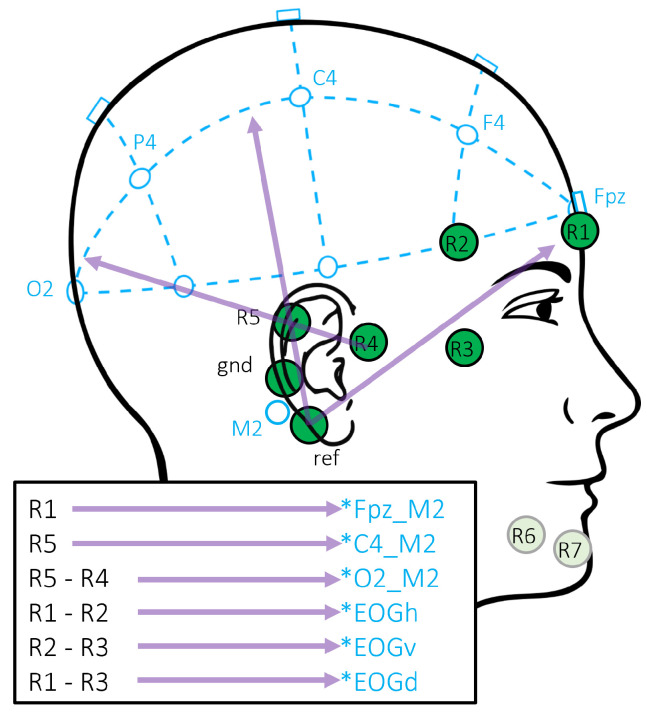
trEEGrid channel positions and their recombinations. Re-referenced channels approximate classical PSG positions, as well as EOG and EMG, marked with an asterisk (*) to indicate the approximation.

**Figure 3 diagnostics-14-00909-f003:**
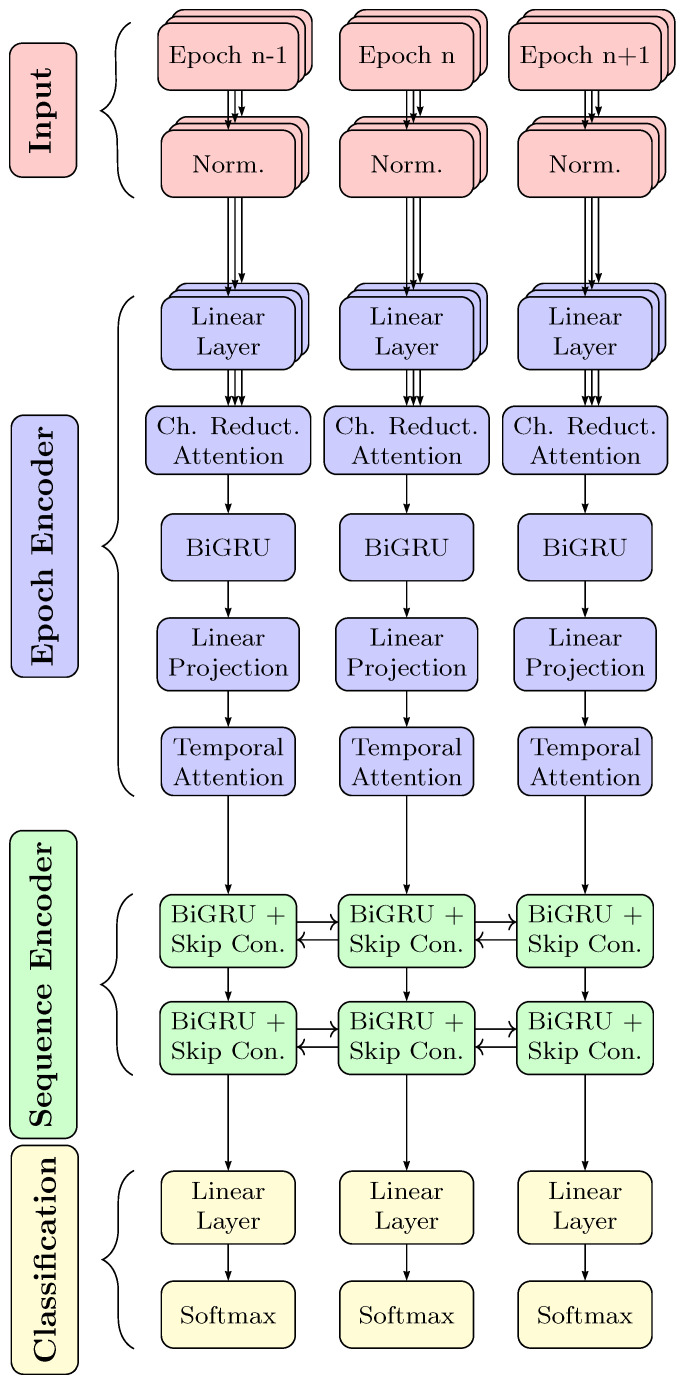
Overview of the employed neural network RobustSleepNet based on [30]. The center column shows the processing steps of epoch *n*. Building blocks left and right at epochs n−1 and n+1 share the same weights and are shown to show the inter-epoch dependency processing. As other sequence-to-sequence networks used for sleep staging, the network consists of an epoch encoder, sequence encoder and classifier. In the RobustSleepNet, the epoch encoder is capable of working with an arbitrary number of input channels. Depicted by the layered boxes, the input channels are processed independently first. An attention block merges the input channels for further processing. Finally, the softmax output gives the prediction of the sleep stage.

**Figure 4 diagnostics-14-00909-f004:**
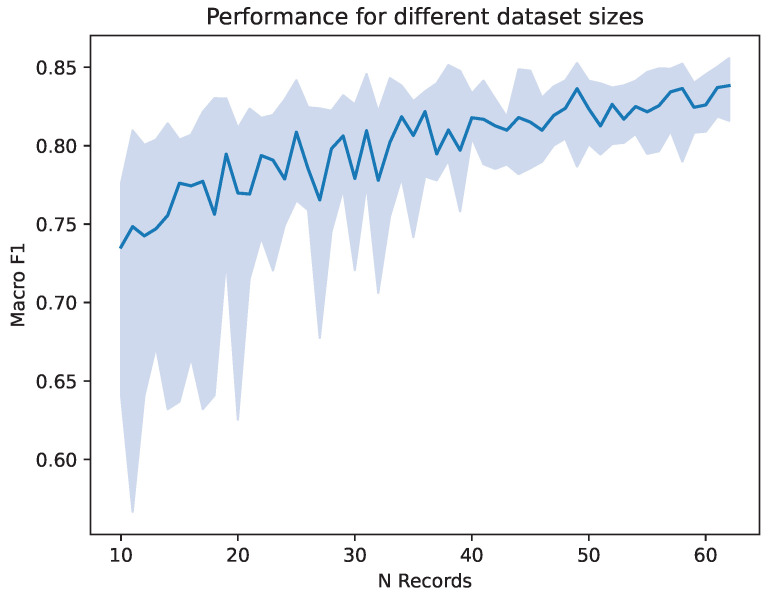
Macro-$F_{1}$ score achieved on different subsets of MASS SS3. The number of records drawn from MASS SS3 is listed on the x axis. Shown are the median and the quarter percentiles of the macro-$F_{1}$ score, over 20 repeated runs on randomly drawn subsets. Calculations were carried out with the SS3^grid-PSG^ setup.

**Table 1 diagnostics-14-00909-t001:** Selection of channels used for each channel setup. In the case of the two trEEGrid setups, re-referenced channels are denoted as the difference of two channels.

Label	EEG Channels	EOG Channels
SS3	all 20	all 2
SS3^grid^	Fz, F8, T4	Right Horiz
SS3^grid-PSG^	(Fp1 + Fp2)/2, C4, O2	Right Horiz
trEEGrid	R1, R2, R4, R5	R1-R3
trEEGrid^PSG^	R1, R5, R5-R4	R1-R3

**Table 2 diagnostics-14-00909-t002:** Parameters used for cross validation setup for both datasets.

	MASS SS3	trEEGrid
number of recordings	62	12
number of folds	10	12
training	70%	9 recordings
validation	20%	2 recordings
test	10%	1 recording

**Table 3 diagnostics-14-00909-t003:** Performance measurements for Learning From Scratch (LFS), Direct Transfer (DT), Finetuning (FT), for the considered dataset and channels setups. Shown are the κ and macro-$F_{1}$ scores over all classes, as well as $F_{1}$ scores broken down for each class and their standard deviations. The best scores for each training type (LFS, DT, FT) are marked in bold.

Type	Source Domain	Target Domain	*κ*	$MF_{1}$	$F_{1,W}$	$F_{1,N1}$	$F_{1,N2}$	$F_{1,N3}$	$F_{1,\mathrm{REM}}$
LFS	SS3	SS3	0.82±0.02	0.84±0.02	0.91±0.05	0.65±0.05	0.91±0.01	0.82±0.04	0.90±0.02
LFS	SS3^grid^	SS3^grid^	0.80±0.02	0.82±0.02	0.89±0.04	0.61±0.02	0.90±0.02	0.80±0.06	0.88±0.03
LFS	SS3^grid-PSG^	SS3^grid-PSG^	0.78±0.02	0.81±0.02	0.87±0.03	0.61±0.06	0.89±0.01	0.79±0.04	0.89±0.02
LFS	trEEGrid	trEEGrid	0.63±0.14	0.66±0.10	0.70±0.12	0.46±0.15	0.73±0.09	0.77±0.20	0.66±0.25
LFS	trEEGrid^PSG^	trEEGrid^PSG^	0.74±0.05	0.77±0.04	0.77±0.11	0.60±0.10	0.76±0.07	0.87±0.04	0.85±0.08
DT	SS3	trEEGrid	0.56±0.14	0.63±0.10	0.74±0.14	0.48±0.11	0.71±0.11	0.66±0.21	0.56±0.24
DT	SS3	trEEGrid^PSG^	0.57±0.11	0.63±0.08	0.72±0.13	0.47±0.11	0.71±0.11	0.68±0.18	0.60±0.20
DT	SS3^grid^	trEEGrid	0.59±0.14	0.63±0.11	0.73±0.15	0.41±0.14	0.72±0.10	0.68±0.21	0.62±0.24
DT	SS3^grid^	trEEGrid^PSG^	0.60±0.12	0.64±0.11	0.71±0.13	0.40±0.14	0.72±0.11	0.71±0.19	0.65±0.27
DT	SS3^grid-PSG^	trEEGrid	0.62±0.14	0.64±0.10	0.67±0.17	0.34±0.12	0.73±0.11	0.71±0.21	0.76±0.11
DT	SS3^grid-PSG^	trEEGrid^PSG^	0.62±0.16	0.65±0.13	0.71±0.14	0.35±0.16	0.71±0.13	0.74±0.18	0.73±0.20
FT	SS3	trEEGrid	0.71±0.15	0.74±0.12	0.77±0.08	0.59±0.15	0.80±0.07	0.75±0.19	0.80±0.19
FT	SS3	trEEGrid^PSG^	0.70±0.12	0.75±0.08	0.85±0.06	0.58±0.08	0.77±0.10	0.80±0.11	0.76±0.19
FT	SS3^grid^	trEEGrid	0.69±0.13	0.75±0.10	0.80±0.09	0.61±0.15	0.77±0.10	0.77±0.17	0.79±0.17
FT	SS3^grid^	trEEGrid^PSG^	0.78±0.09	0.81±0.05	0.84±0.11	0.62±0.09	0.81±0.10	0.87±0.06	0.88±0.07
FT	SS3^grid-PSG^	trEEGrid	0.70±0.17	0.73±0.12	0.79±0.14	0.51±0.14	0.80±0.10	0.77±0.21	0.79±0.18
FT	SS3^grid-PSG^	trEEGrid^PSG^	0.76±0.09	0.79±0.04	0.86±0.06	0.61±0.13	0.79±0.10	0.83±0.07	0.88±0.08
Scorer [6]	-	SIESTA [36]	0.76	-	-	-	-	-	-
Our Scorer [9]	-	trEEGrid^PSG^	0.70	0.74	0.78	0.51	0.78	0.85	0.80

## Data Availability

The availability of the data presented in this study is limited due to General Data Protection Regulations (GDPR).

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
