# Peer review of "Transfer Learning for Automatic Sleep Staging Using a Pre-Gelled Electrode Grid"

_diagnostics, 2024, doi:10.3390/diagnostics14090909_

Round 1

Reviewer 1 Report

Comments and Suggestions for Authors

Good approach. Please adjust the following points below:

1. Introduction part: Add more discussion on Transfer Learning for Automatic Sleep Staging.

2. Method: Did you check with different Epochs and how was the variation?

How did you optimize your model?

3. Discussion: Please discuss with other relevant articles. What are the strengths and weaknesses of your study?

4. Please add a conclusion. separately.

Author Response

Dear Reviewer,
Thank you very much for taking the time to review this manuscript. Please find the detailed responses in the attachment.

Reviewer 2 Report

Comments and Suggestions for Authors

Regarding the manuscript with the title

"Transfer Learning for Automatic Sleep Staging Using a Pre-Gelled Electrode Grid"

where in the authors proposed new sensor solutions for home sleep monitoring to improve sleep medicine care and enable long-term observation. They emphasized the need for automated assessment due to differences in sensor data compared to traditional polysomnography. To deal with limited training data for new sensor technology, the author used pre-training on large datasets and fine-tuning on small datasets. By pre-training a publicly available polysomnography dataset and fine-tuning the new sensor data, they achieved an F1 score of 0.81 for automatic sleep phase detection using EEG and EOG signals. The analysis also included considerations of spatial distribution and approximation of classical electrode positions based on linear combinations of new sensor network channels.

This study seems to be well written. The content is well organized. This study has covered an existing gap in the existing field. It seems to be valuable and has high scientific value. But in my opinion, as I reviewed it several times, it needs a minor revision.

It is better for the authors to fix these problems and improve its quality.

1: The abstract is extremely well written. But it seems that interdiction suffers from the lack of a proper structure and logical writing routine. I suggest to rewrite the introduction in this way, first from the problems and challenges

Write automatic sleep staging to justify your study, then discuss artificial intelligence and its applications in health care. I suggest that in order to familiarize the audience and researchers, as well as to increase the readability of your study, you must mention the various applications of artificial intelligence in the field of health care. For this purpose, you can use these studies.

"Unsupervised Domain Adaptation of MRI Skull-Stripping Trained on Adult Data to Newborns"

"An Intelligent Modular Real-Time Vision-Based

System for Environment Perception"

"Automated detection model in classification of B-lymphoblast cells from normal B-lymphoid precursors in blood smear microscopic images based on the majority voting technique"

   The use of these sources is optional, but it can provide a variety of applications of artificial intelligence and machine learning in the field of health care for your manuscript.

2: Figure four has no caption and this is confusing for me the reviewer and even the readers of this article. So please mention a suitable jacket for it.

3: Regarding the implementation code, it is better to share it so that the reviewer can understand the correctness of your implementation.

In the end, I must inform you that this manuscript is valuable and has the ability to be published in this journal.

Author Response

(The authors gave the same response as above.)

Reviewer 3 Report

Comments and Suggestions for Authors

The main question that is addressed in the present research is the automatic sleep phase detection with the use of pre-training and new sensor technology approaches, taking into consideration that sleep laboratory and the measurement setup can have a bad influence on the patients sleep.

2. The original part in the filed is the use of novel sensors. The specific gap in the field that this work is addressing is the use methods that are easily applicable and allows EEG sleep monitoring at home at a high quality.

3. Every innovative approach in the field is wellcomed and this one used network architecture, for transfer learning and applied  on a smaller dataset to pre-train  a prototype of a self-applicable electrode grid (easy for patient use) capable of recording EEG, EOG and EMG at home. 

4. All experiments deal with normal people and this work could be considered including specific desease conditions even as pilot study strengthening the applicability of the approach. Authprs al least could discuss possible problems that could face their innovation in real patients. 

5. Conclusions are consistent with the evidence and arguments presented. .

6. The references are appropriate as well as tables and figures and

quality of the data.

Author Response

Dear Reviewer,
Thank you very much for taking the time to review this manuscript. Please find the detailed responses attached.

Round 2

Reviewer 1 Report

Comments and Suggestions for Authors

Great effort. I have no more comments.